# *BTK*, *NUTM2A,* and *PRPF19* Are Novel *KMT2A* Partner Genes in Childhood Acute Leukemia

**DOI:** 10.3390/biomedicines9080924

**Published:** 2021-07-30

**Authors:** Elena Zerkalenkova, Svetlana Lebedeva, Aleksandra Borkovskaia, Olga Soldatkina, Olga Plekhanova, Grigory Tsaur, Michael Maschan, Aleksey Maschan, Galina Novichkova, Yulia Olshanskaya

**Affiliations:** 1Dmitry Rogachev National Medical Research Center of Pediatric Hematology, Oncology and Immunology, 117997 Moscow, Russia; svetlana.lebedeva@fccho-moscow.ru (S.L.); aleksandra.borkovskaia@fccho-moscow.ru (A.B.); olga.soldatkina@fccho-moscow.ru (O.S.); Michael.Maschan@fccho-moscow.ru (M.M.); Aleksey.Maschan@fccho-moscow.ru (A.M.); Galina.Novichkova@fccho-moscow.ru (G.N.); Yulia.Olshanskaya@fccho-moscow.ru (Y.O.); 2Regional Children Hospital 1, Pediatric Oncology and Hematology Center, Research Institute of Medical Cell Technologies, Ural Federal University Named after the First President of Russia BN Yeltsin, 620149 Ekaterinburg, Russia; elalmatveeva@gmail.com (O.P.); grigory.tsaur@gmail.com (G.T.)

**Keywords:** pediatric acute leukemia, fusion genes, *KMT2A*, cytogenetics, NGS

## Abstract

Chromosomal rearrangements of the human *KMT2A*/*MLL* gene are associated with acute leukemias, especially in infants. *KMT2A* is rearranged with a big variety of partner genes and in multiple breakpoint locations. Detection of all types of *KMT2A* rearrangements is an essential part of acute leukemia initial diagnostics and follow-up, as it has a strong impact on the patients’ outcome. Due to their high heterogeneity, *KMT2A* rearrangements are most effectively uncovered by next-generation sequencing (NGS), which, however, requires a thorough prescreening by cytogenetics. Here, we aimed to characterize uncommon *KMT2A* rearrangements in childhood acute leukemia by conventional karyotyping, FISH, and targeted NGS on both DNA and RNA level with subsequent validation. As a result of this comprehensive approach, three novel *KMT2A* rearrangements were discovered: ins(X;11)(q26;q13q25)/*KMT2A-BTK*, t(10;11)(q22;q23.3)/*KMT2A-NUTM2A*, and inv(11)(q12.2q23.3)/*KMT2A-PRPF19*. These novel *KMT2A*-chimeric genes expand our knowledge of the mechanisms of *KMT2A*-associated leukemogenesis and allow tracing the dynamics of minimal residual disease in the given patients.

## 1. Introduction

Over the recent years, the novel molecular techniques have substantially changed the face of clinical laboratory diagnostics. However, in certain types of hematological malignancies, the cytogenetic studies are still mandatory for classification and risk stratification [1]. They uncover most significant chromosomal aberrations in a fast and cost-effective way, selecting peculiar cases for the advanced molecular approach.

*KMT2A* (histone-lysine N-methyltransferase 2A, former *MLL*)-rearranged acute leukemia represents one type of such neoplasms that are complex both for diagnostics and treatment. They are commonly found in infant acute lymphoblastic leukemia (ALL)—up to 70–80% [2,3], and with less frequency in ALL or acute myeloid leukemia (AML) in a broader age range of older children or adults [4,5]. In pediatric AML, the incidence of *KMT2A* rearrangements is approximately 15–25% [6,7,8]. *KMT2A* rearrangements are an important predictor of outcome in acute leukemia, with the prognosis varying from good and intermediate to poor depending on the partner gene and breakpoint location [2,9,10]. Therefore, the accurate detection of all *KMT2A* rearrangement types is crucial to correct the therapy and risk groups definition.

The diagnostic struggle lies in a huge heterogeneity of *KMT2A* gene rearrangement. It involves multiple partner genes—at least 94 forward and 247 reciprocal partners were summarized in the recent recombinome [8]. Six most frequent partner genes (*AFF1*, *MLLT3*, *MLLT10*, *MLLT1*, *ELL*, *AFDN*) are found in about 80% of *KMT2A*-positive acute leukemias, others are rarer, and for most partner genes only single cases are described [8,11]. The main *KMT2A* partners (*AFF1*, *MLLT3*, *MLLT1*, *MLLT10*, *ELL*) encode the factors controlling transcriptional elongation by RNA polymerase II. Other gene partners are responsible for various functions in a cell such as cellular signaling, scaffolding, mitochondrial and cytosolic enzymatic activity, etc. [12,13]. In addition, *KMT2A* has multiple breakpoint locations. Most rearrangements occur within the major breakpoint cluster (introns 8 to 14), but cases involving DNA upstream as well as downstream are known [8]. Systematic investigation of these outliers has led to the discovery of the minor breakpoint cluster (introns 19 to 24) [14]. This molecular diversity challenges the initial molecular diagnostics and the monitoring of minimal residual disease (MRD) in acute leukemia harboring *KMT2A* rearrangement.

The evaluation of all types of *KMT2A* rearrangements used to require labor-intensive methods such as long-distance inverse PCR [15,16]. However, the implementation of the next-generation sequencing (NGS) enables an easier detection of *KMT2A* rearrangements on both the DNA and RNA levels [16]. A custom NGS panel of capture probes covering the whole *KMT2A* gene has demonstrated a great efficacy in DNA breakpoints analysis including the minor breakpoint cluster reveal [14]. Another useful NGS strategy is the anchored multiplex PCR target enrichment that detects fusion transcripts when one partner gene (e.g., *KMT2A*) is known [17]. These methods are robust and convenient but expensive. They demand a thorough prescreening, which is perfectly accomplished by cytogenetics.

In this study, we report three cases of novel *KMT2A* partner genes uncovered as a result of comprehensive cytogenetic and molecular genetic approach—Bruton’s tyrosine kinase (*BTK*) in infant AML with (X;11)(q22.1;q23.3), NUT family member 2A (*NUTM2A*) in pediatric secondary T-cell ALL with t(10;11)(q22;q23.3), and pre-mRNA processing factor 19 (*PRPF19*) in infant AML with inv(11)(q12.2q23.3). The *KMT2A* gene rearrangements were seen in karyotype, confirmed by break-apart FISH. The common rearrangements were excluded by translocational FISH and RT-PCR, and the samples were headed to targeted NGS.

## 2. Materials and Methods

### 2.1. Patient 1

A 9 m.o. girl with a month history of hyperleukocytosis was admitted to the hospital’s Onco-hematology department. Upon admission, the patient demonstrated hepatomegaly (3 cm below the costal margin) and splenomegaly (1 cm below the costal margin). No signs of lymphadenopathy were found. Blood tests revealed a white blood cells count of 63.18 × 10^9^ /L containing 2% blasts, hemoglobin 10.2 g/dL, platelet count of 234 × 10^9^ /L, and red blood cells count of 3.87 × 10^12^ /L. Morphological and immunological examinations revealed the diagnosis of acute monocytic leukemia (M5b according to the French-American-British (FAB) classification system). The patient was treated according to the local protocol AML-MRD-2018. The induction treatment consisting of Ara-C (cytosine arabinoside), mitoxantrone, and etoposide (AM42E block) was performed and a complete MRD-negative remission by flow cytometry and RT-PCR was achieved. The consolidation treatment consisting of high-dose Ara-C and mitoxantrone (HAM block) was complicated by myelosuppression and the development of septic shock, which in turn led to the multiple organ failure and patient’s death.

### 2.2. Patient 2

An 8 y.o. girl with a history of teratoid-rhabdoid tumor 4 years prior was admitted to the hospital Onco-hematology department with pleural effusion suspected of lymphoma. The patient was diagnosed with T-cell acute lymphoblastic leukemia (ALL TIII) with 80% CD1a + CD2 + CD3 + CD4 + CD7 + CD8 + TCRab + CD45 + CD3cyt+ blasts by flow cytometry. The patient was treated according to the ALL IC BFM 2009 protocol. The patient reached MRD-negative remission by RT-PCR by the end of induction course and then was lost for follow-up.

### 2.3. Patient 3

A 10 m.o. girl was admitted to the hospital due to febrile fever. Upon admission, the patient demonstrated hepatomegaly (3 cm below the costal margin) and splenomegaly (3 cm below the costal margin). The blood tests revealed a white blood cells count of 10.8 × 10^9^/L containing 5% blasts, hemoglobin 5.3 g/dL, platelet count of 53 × 10^9^/L, and red blood cells count of 1.84 × 10^12^/L. Morphological and immunological examinations revealed the diagnosis of acute myelomonocytic leukemia (FAB M4). The patient was treated according to the local protocol AML-MRD-2018. The induction treatment consisting of Ara-C (cytosine arabinoside), mitoxantrone, and etoposide (AM42E block) was performed and a complete MRD-negative remission by flow cytometry and RT-PCR was achieved. The patient is now receiving a consolidation treatment consisting of a high-dose of Ara-C and mitoxantrone.

All the patients underwent comprehensive cytogenetic and molecular genetic diagnostics following the identical karyotyping—FISH—PCR—NGS—validation pipeline. Informed consent for the diagnostic work-up was obtained from the legal guardians of the patients.

A bone marrow aspirate obtained at diagnosis was cultured overnight without mitogenic stimulation and processed as previously described [18]. G-banded karyotyping was performed according to An International System for Human Cytogenomic Nomenclature [19]. The *KMT2A* gene rearrangement was confirmed by FISH with the Kreatech ON *KMT2A* break-apart probe (Leica Microsystems B.V., Amsterdam, Netherlands). The particular types of *KMT2A* rearrangements were excluded by FISH with translocational probes Kreatech ON KMT2A/AFF1, ON KMT2A/AFDN, ON KMT2A/MLLT1, ON KMT2A/MLLT3 (Leica Microsystems B.V., Amsterdam, The Netherlands), and CytoCell KMT2A/MLLT10 (CytoCell, Oxfordshire, United Kingdom). All FISH procedures were performed according to the manufacturers’ instructions, 200 interphase nuclei and 20 metaphase plates were analyzed for each probe.

Total DNA and RNA were simultaneously extracted from the whole bone marrow samples using the InnuPrep DNA/RNA Mini Kit (Analytik Jena AG, Jena, Germany). Eight most common *KMT2A* fusions were excluded by the reverse transcription-polymerase chain reaction (RT-PCR) with forward *KMT2A* primers and TaqMan probes [20] with reverse *EPS15* [21], *AFF1*, *MLLT3*, *MLLT1* [20], *AFDN* [22], *MLLT10* [23], *MLLT6* [24], and *ELL* [25] primers. To detect unknown *KMT2A* fusion transcripts, NGS with anchored multiplex PCR target enrichment by the FusionPlex Myeloid panel (ArcherDX, Boulder, CO, USA) was used. DNA breakpoints were analyzed with the *KMT2A*-targeted custom NGS panel [14]. NGS was carried out on Illumina MiSeq (Illumina, San Diego, CA, USA). Breakpoint junctions on DNA and RNA levels were validated by Sanger sequencing. The achieved data on the *KMT2A* translocation partner and breakpoint location was used for patient-specific MRD monitoring.

## 3. Results

### 3.1. KMT2A-BTK Fusion Gene in AML with ins(X;11)(q26;q13q25)

Conventional cytogenetics and FISH of bone marrow aspirates showed 45,X,ins(X;11)(q26;q13q25),−20 karyotype with 95% *KMT2A*-rearranged nuclei (Figure 1a,b). The eight most frequent translocations with partner genes *EPS15*, *AFF1*, *MLLT3*, *MLLT1*, *AFDN*, *MLLT10*, *MLLT6,* and *ELL* were excluded by FISH with corresponding translocation probes and multiplex RT-PCR screening.

The analysis by NGS with anchored multiplex PCR target enrichment revealed novel KMT2A-BTK fusion transcripts with exon 9-exon 2 and exon 8-exon 2 breakpoint junctions. RT-PCR with *KMT2A* exon 8 and exon 9 forward and *BTK* exon 2 reverse primers confirmed the expression of both fusion transcripts with the latter being non-functional due to the premature stop codon. The *KMT2A*-targeted NGS panel showed the *KMT2A-BTK* fusion gene with intron 9-intron 1 junction and non-template insert in between. The sequences were validated by the Sanger method (Figure 1c,d) and submitted to the GenBank database. Accession numbers and validation primers are summarized in Appendix A. RNA validation primers were used for RT-PCR-based MRD monitoring.

### 3.2. KMT2A-NUTM2A Fusion Gene in T-ALL with t(10;11)(q22;q23.3)

Bone marrow and pleural fluid aspirates were analyzed by G-banded karyotyping and FISH with the *KMT2A* break-apart probe. Pleural fluid aspirates were also analyzed by real-time RT-PCR for the eight most common *KMT2A* rearrangements screening and headed to the NGS analysis. Conventional cytogenetics and FISH showed 47,XX,t(10;11)(q22;q23),+mar karyotype with 100% *KMT2A*-rearranged nuclei (Figure 2a,b). The *KMT2A* break-apart probe demonstrated an unusual signal distribution with the additional copy of rearranged 3′-portion of *KMT2A* gene, which was placed on the marker chromosome as shown by metaphase FISH (Appendix A). Common rearrangements were excluded. The NGS with anchored multiplex PCR target enrichment identified the novel KMT2A-NUTM2A fusion transcript with exon 11-exon 1 breakpoint junction, which was Sanger validated (Figure 2d, Table 1).

Genomic DNA analysis revealed the junction of *KMT2A* intron 11 to *NUTM2A* upstream of DNA designating the so-called spliced fusion. The sequences of the forward (Figure 2c) and reverse chimeric genes were confirmed by direct PCR from genomic DNA followed by Sanger sequencing and placed in the GenBank database (Table 1; see Appendix A for validation primers). RNA validation primers were used for RT-PCR-based MRD monitoring.

### 3.3. KMT2A-PRPF19 Fusion Gene in AML with inv(11)(q12.2q23.3)

Conventional cytogenetics and FISH of bone marrow aspirates showed 46,XX,inv(11)(q12.2q23.3) [20] karyotype (Figure 3a) with 90% *KMT2A*-rearranged nuclei. Separate 5′ and 3′ *KMT2A* probes were both located within the 11q chromosomal band, indicating the inversion of this chromosome (Figure 3b). The sample was immediately headed to the *KMT2A*-targeted NGS panel, which revealed a novel *KMT2A-PRPF19* fusion gene with intron 8-intron 1 junction. RT-PCR with *KMT2A* exon 8 forward and *PRPF19* exon 2 reverse primers confirmed the expression of the corresponding fusion transcript. The sequences were validated by the Sanger method (Figure 3c,d) and submitted to the GenBank database (Table 1; see Appendix A for validation primers). Validation primers are summarized in Appendix A. RNA validation primers were used for RT-PCR-based MRD monitoring.

## 4. Discussion

As a result of karyotyping—FISH—PCR—NGS—validation comprehensive diagnostic work-up, three novel *KMT2A* rearrangements in childhood acute leukemia were discovered: ins(X;11)(q26;q13q25)/*KMT2A-BTK*, t(10;11)(q22;q23.3)/*KMT2A-NUTM2A*, and inv(11)(q12.2q23.3)/*KMT2A-PRPF19*.

To the best of our knowledge, the presented ins(X;11)(q26;q13q25) infant AML is the first case of *BTK* translocations in hematologic malignancies, and its value for leukemogenesis is to be further investigated. However, as the breakpoint junction is located in *BTK* intron 1, upstream of its coding region (the translation starts in exon 2), we can speculate that BTK structure itself is not committed and its activity is increased due to the upregulated expression together with KMT2A.

The *BTK* gene located in the chromosomal band Xq22.1 encodes for a Tec family kinase—Bruton tyrosine kinase. This protein is known to play a crucial role in B-cell maturation [26,27]. *BTK* mutations cause X-linked agammaglobulinemia type 1, which is associated with a failure of Ig heavy chain rearrangement [28]. In addition to the role in normal B-cells development, the BTK high expression has long been known in B-cell malignancies (e.g., multiple myeloma, mantle cell lymphoma, and chronic lymphocytic leukemia) [29,30,31]. BTK also acts as a tumor suppressor in other cells, supposedly, due to its ability to stabilize the p53 protein and increase the p73 protein level [32].

*BTK* expression is found in all hematopoietic lineages except in T-cells [33,34]. However, in contrast to lymphoid maturation, BTK signaling in healthy myelopoiesis as well as the development of AML is poorly understood. Many studies have shown that AML cell lines and primary AML cultures are characterized by increased expression and constitutive activity of BTK [33,35]. According to several studies, BTK inhibition leads to a significant reduction in cell expansion potential in both primary AML cultures and AML cell lines [33,36]. This suggested the role of BTK as a potential therapeutic target in acute myeloid leukemia, so preclinical and clinical studies are performed to estimate the BTK selective inhibitor ibrutinib usage in the treatment of AML [33,36,37,38].

To date, three BTK inhibitors are approved by FDA: ibrutinib, acalabrutinib, and zanubrutinib. Several preclinical and clinical studies were performed to determine whether BTK inhibitors could be used in the treatment of AML. The first-in-class BTK inhibitor, ibrutinib, was shown to reduce AML blasts proliferation and enhance the cytotoxic effect in combination with chemotherapy [33,36,37]. Additional preclinical studies demonstrated the activity of ibrutinib in AML with internal tandem duplication of the juxtamembrane region of the FMS-like tyrosine kinase 3 receptor (FLT3-ITD) [38]. It was also shown that the new BTK inhibitor, abivertinib (AC0010), inhibits cell proliferation and reduces the colony forming capacity of AML cells, especially harboring FLT3-ITD mutation, by BTK-dependent and independent mechanisms [39]. The open-label phase 2a multicenter clinical study, which evaluates the efficacy and safety of ibrutinib alone or in combination with either cytarabine or azacitidine in adult patients with AML (for whom the standard treatment had failed or who had not received prior therapy and who refused standard treatment options) was terminated due to limited efficacy. Apparently, the use of ibrutinib alone or in combination with chemotherapy in AML does not improve the treatment outcomes [40]. Clinical studies on the efficacy and safety of abivertinib in patients with AML are planned.

The t(10;11)(q22;q23.3)/*KMT2A-NUTM2A* chromosomal translocation in pediatric T-ALL in our study is the first case demonstrating the *NUTM2A* rearrangement in hematological malignancies. This gene and its highly similar homolog *NUTM2B* have previously been identified in endometrial stromal sarcomas forming in-frame fusion with *YWHAE* [41] and small round cell sarcomas forming in-frame fusion with *CIC* [42]. Our study showed fusion formation between *KMT2A* intron 11 and *NUTM2A* upstream DNA, which results in the so-called spliced fusion. In this type of *KMT2A* rearrangement, the partner breakpoint is located in genomic DNA upstream of the gene itself [43]. This phenomenon is especially common in t(11;19)(p13;q23)/*KMT2A-MLLT1* translocation (for up to half of the cases, see [8]), but it is also known in other rearrangements.

Unlike the two above-mentioned genes, *PRPF19* to our knowledge has never been shown to be involved into cancerogenesis. This is a gene in the 11q12.2 chromosomal band that encodes core ubiquitin-protein ligase involved in pre-mRNA splicing and DNA repair. It acts as a component of spliceosome and is required for spliceosome assembly, stability, and activity [44,45,46,47,48]. PRPF19 can also be recruited to the RNA polymerase II C-terminal domain, thus coupling transcriptional and spliceosomal machineries [49]. In addition to its role in transcriptional processes, PRPF19 is involved in DNA damage response and DNA repair. As a part of the XAB2 complex, PRPF19 is involved in RNA II polymerase transcription stop in sites of DNA damage [50]. PRPF19 can be recruited to the sites of DNA damage and allows the activation of ATR, a master regulator of the DNA damage response [51]. It also plays a role in DNA double-strand break repair by recruiting the repair factor SETMAR to the altered DNA [52]. The above-mentioned features suggest the menace of any *PRPF19* gene alterations for cell integrity. However, only a minor PRPF19 part of the six first amino acids is lost due to the inv(11)(q12.2q23.3) event.

Therefore, we demonstrate here the effective cooperation of conventional cytogenetics and advanced molecular approach for *KMT2A* rearrangements detection. Through this approach, we have discovered three novel *KMT2A* translocation partners in childhood acute leukemia—genes encoding tyrosine kinase BTK, ubiquitin-protein ligase PRPF19, and *NUTM2A*—gene of poorly known function. Studying these new *KMT2A* translocations will provide a better understanding of the molecular mechanisms behind their oncogenic properties. The achieved sequences for novel *KMT2A* fusions are highly leukemia-specific and patient-specific markers for PCR-based MRD monitoring.

## Figures and Tables

**Figure 1 biomedicines-09-00924-f001:**
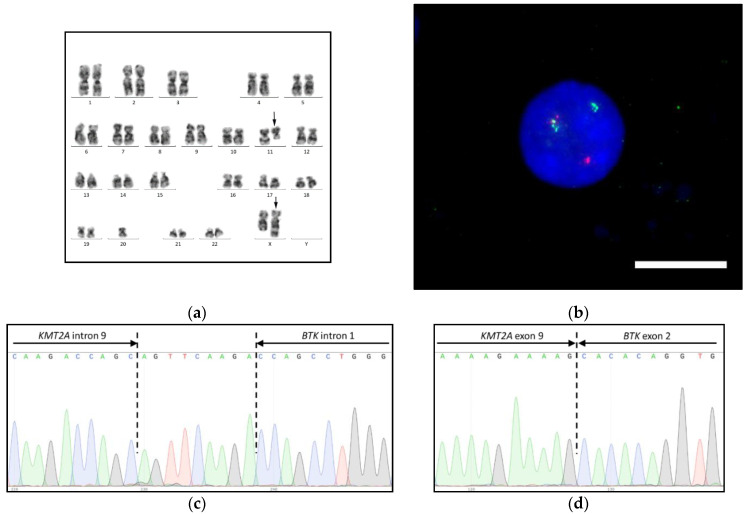
*KMT2A-BTK* fusion gene in AML with ins(X;11)(q26;q13q25): (**a**) 45,X,ins(X;11)(q26;q13q25),−20 karyotype by G-banding, rearranged chromosomes 11 and X are marked with arrows; (**b**) *KMT2A* gene rearrangement by interphase FISH with the Kreatech ON KMT2A break-apart probe (Leica), the bar is 10 μm; (**c**) *KMT2A-BTK* fusion gene; and (**d**) KMT2A-BTK fusion transcript by Sanger sequencing.

**Figure 2 biomedicines-09-00924-f002:**
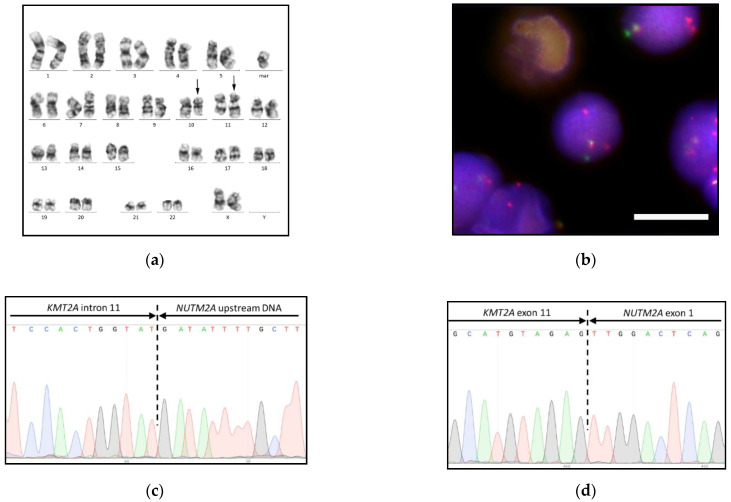
*KMT2A-NUTM2A* fusion gene in T-ALL with t(10;11)(q22;q23.3): (**a**) 47,XX,t(10;11)(q22;q23),+mar karyotype by G-banding, rearranged chromosomes 10 and 11 are marked with arrows; (**b**) *KMT2A* gene rearrangement by FISH with the Kreatech ON KMT2A break-apart probe (Leica), the bar is 10 μm; (**c**) *KMT2A-NUTM2A* fusion gene; and (**d**) KMT2A-NUTM2A fusion transcript by Sanger sequencing.

**Figure 3 biomedicines-09-00924-f003:**
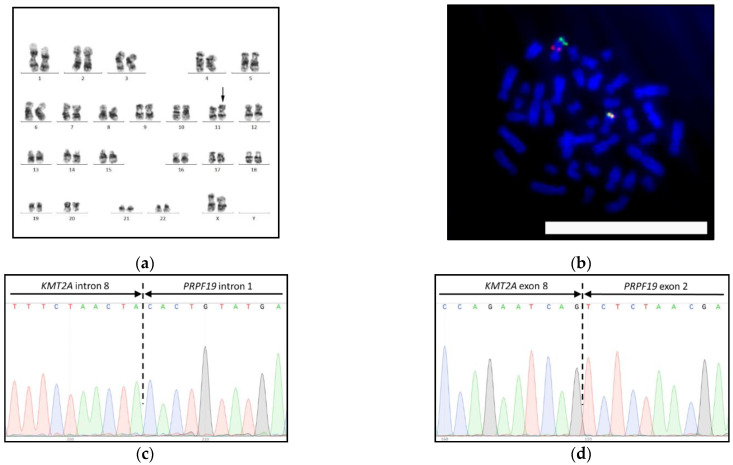
*KMT2A-PRPF19* fusion gene in AML with inv(11)(q12.2q23.3): (**a**) 46,XX,inv(11)(q12.2q23.3) karyotype by G-banding, inverted chromosome 11 is marked with arrows; (**b**) *KMT2A* gene rearrangement by metaphase FISH with the Kreatech ON KMT2A break-apart probe (Leica), the bar is 10 μm; (**c**) *KMT2A-PRPF19* fusion gene; and (**d**) KMT2A-fusion transcript by Sanger sequencing.

**Table 1 biomedicines-09-00924-t001:** GenBank (http://www.ncbi.nlm.nih.gov/Genbank, accessed on 27 July 2021) database accession numbers for novel *KMT2A* fusion genes and transcripts.

Rearrangement	Fusion Gene	Fusion Transcript
ins(X;11)(q26;q13q25)/*KMT2A-BTK*	MN687943	MN238630 (exon 9-exon 2)
t(10;11)(q22;q23.3)/*KMT2A-NUTM2A*	MT721855 (forward) MT721856 (reciprocal)	MT721854
inv(11)(q12.2q23.3)/*KMT2A-PRPF19*	MZ443566	MZ443565

## Data Availability

Data are contained within the article or supplementary material. The sequences of novel *KMT2A* fusion genes and transcripts can be found in the GenBank database (http://www.ncbi.nlm.nih.gov/Genbank, accessed on 27 July 2021) under the accession numbers listed in Table 1.

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
