# Peer review of "BTK, NUTM2A, and PRPF19 Are Novel KMT2A Partner Genes in Childhood Acute Leukemia"

_biomedicines, 2021, doi:10.3390/biomedicines9080924_

Round 1

Reviewer 1 Report

The present work describes a discovery of three novel KMT2A (histone-lysine N-methyltransferaze 2A, former MLL) translocation partners in childhood acute leukaemia. Translocations are important predictors of the acute leukaemia outcome but also this information might provide a better understanding of the mechanisms of leukemogenesis.

The work is very well written and comprehensive, the figures are clear and the methods combine conventional cytogenetics with a modern molecular approach.  The methods are well described and cost-effective, therefore reproducible even in moderately sophisticated laboratories. The discussion is well structured, the literature is novel and up to date.

I congratulate the authors for their discovery.

Author Response

The authors sincerely thank the reviewer for the nice response.

Reviewer 2 Report

In the present paper, authors report here three cases of novel KMT2A partner genes uncovered as a result of comprehensive cytogenetic and molecular genetic approach – Bruton’s tyrosine kinase (BTK) in infant AML with (X;11)(q22.1;q23.3), NUT family member 2A (NUTM2A) in pediatric secondary T-cell ALL with t(10;11)(q22;q23.3) and pre-mRNA processing factor 19 (PRPF19) in infant AML with inv(11)(q12.2q23.3). KMT2A gene rearrangements were seen in karyotype, confirmed by break-apart FISH, common rearrangements were excluded by translocational FISH and RT-PCR, and the samples were headed to targeted NGS.

The report of three novel chimeric KMT2/partners will be interesting and have a role for pediatric hematology oncology. I think this report is adequate for the journal.

Comments:

  1. About patient 1, both MN687943 and MN238630 are partial sequence. If authors studied the entire sequence of KMT2A-BTK Fusion cDNA, they should submit it the database because this is the first report of the chimera. As they mentioned in Discussion, BTM seems to work as oncogenic in AML. Therefore, functional BTK expression in the patient’s blast will be informative for readers.

  1. About patient 2, I could not get the record of MT721854 in GenBank. It will be opened after acceptance?

  1. About patient 3, I could not get the record of MZ443565 in GenBank.  It will be opened after acceptance?

  1. Authors mentioned the effect of BTK inhibition in Discussion. Since several BTK inhibitors have been evaluated in clinical studies, it will be informative to introduce those reports in discussion.

Author Response

  1. About patient 1, both MN687943 and MN238630 are partial sequence. If authors studied the entire sequence of KMT2A-BTK Fusion cDNA, they should submit it the database because this is the first report of the chimera. As they mentioned in Discussion, BTM seems to work as oncogenic in AML. Therefore, functional BTK expression in the patient’s blast will be informative for readers.

RESPONSE 1 

We did not study the whole chimeric gene/transcript sequence as both MLL-targeted DNA NGS panel and Archer FusionPlex kit are designed in such a way that they provide only breakpoint junctions, which we in turn validated. Only validated (namely breakpoint junction) sequences were submitted to GenBank.

The authors sincerely thank the reviewer for the point on BTK expression. It was planned, however, no initial material is left available for the BTK protein expression analysis either by immunocytochemistry or by western blot.

We have performed preliminary experiments on BTK RNA expression via RT-PCR. ABL was used as a housekeeping control, dCt was used to estimate the expression of BTK RNA in our sample vs groups of MLL-positive AML as well as FLT3-ITD-positive MLL-negative and FLT3-ITD-negative MLL-negative AMLs. Please see attached the scattergram. At first sight there is no sign of elevated BTK expression within our sample. However, the groups are too small to make any conclusions. Besides, we are still in search for the adequate control group for ddCt quantitative estimation and we are still performing analysis of available RNAseq data of ours. This is a subject of the future research.

  1. About patient 2, I could not get the record of MT721854 in GenBank. It will be opened after acceptance?

RESPONSE 2

GenBank records for the patient 2  are not opened yet. Please find the submission forms attached.

  1. About patient 3, I could not get the record of MZ443565 in GenBank.  It will be opened after acceptance?

RESPONSE 3

GenBank records for the patient 3  are not opened yet. Please find the submission forms attached.

  1. Authors mentioned the effect of BTK inhibition in Discussion. Since several BTK inhibitors have been evaluated in clinical studies, it will be informative to introduce those reports in discussion.

The authors sincerely thank the reviewer for the point on BTK inhibitors. The chapter considering BTK inhibitors is added to the discussion.

Round 2

Reviewer 2 Report

Authors appropriately addressed my questions and comments.